# A Blind Nonlinearity Compensator Using DBSCAN Clustering for Coherent Optical Transmission Systems

**Elias Giacoumidis [1,\*], Yi Lin [1], Mutsam Jarajreh [2], Sean O'Duill [1], Kevin McGuinness [3], Paul F. Whelan [4] and Liam P. Barry [1]**

1   Radio and Optical Laboratory, School of Electronic Engineering, Dublin City University, Glasnevin 9, Dublin, Ireland; yi.lin6@mail.dcu.ie (Y.L.); sean.oduill@dcu.ie (S.O.); liam.barry@dcu.ie (L.P.B.)
2   Computer Engineering Department, Fahad Bin Sultan University, Tabuk 47721, Saudi Arabia; mutsam.jarajreh@gmail.com
3   Insight Centre for Data Analytics, School of Electronic Engineering, Dublin City University, Dublin 9, Ireland; kevin.mcguinness@dcu.ie
4   Vision Systems Group, School of Electronic Engineering, Dublin City University, Dublin 9, Ireland; Paul.Whelan@dcu.ie
\*   Correspondence: elias.giacoumidis@dcu.ie; Tel.: +353-1-700-8598

**Abstract:** Coherent fiber-optic communication systems are limited by the Kerr-induced nonlinearity. Benchmark optical and digital nonlinearity compensation techniques are typically complex and tackle deterministic-induced nonlinearities. However, these techniques ignore the impact of stochastic nonlinear distortions in the network, such as the interaction of fiber nonlinearity with amplified spontaneous emission from optical amplification. Unsupervised machine learning clustering (e.g., K-means) has recently been proposed as a practical approach to the blind compensation of stochastic and deterministic nonlinear distortions. In this work, the Density-Based Spatial Clustering of Applications with Noise (DBSCAN) algorithm is employed, for the first time, for blind nonlinearity compensation. DBSCAN is tested experimentally in a 40 Gb/s 16 quadrature amplitude-modulated system at 50 km of standard single-mode fiber transmission. It is shown that at high launched optical powers, DBSCAN can offer up to 0.83 and 8.84 dB enhancement in Q-factor when compared to conventional K-means clustering and linear equalisation, respectively.

**Keywords:** fiber optics communications; coherent communications; machine learning; clustering; nonlinearity cancellation

## 1. Introduction

Coherent optical communications have been proposed as a viable solution for maximising the signal capacity in both short-reach and long-haul communications [1]. However, Kerr-induced fiber nonlinearity prevents channel capacity from approaching the Shannon limit, especially when the signal power is high [2]. Endeavours to surpass the Kerr nonlinearity limit have been performed by techniques that in principle compensate deterministic nonlinearities [3]. For example, nonlinearities can be combated by either inserting an optical phase conjugator (OPC) at the middle point of the link [4], or by inverting the fiber effects among multiple frequency stabilised optical signals [5]. However, OPC reduces the flexibility in an optically routed network, whereas in [5], a digital back propagation (DBP) [6] pre-compensator was used, which is of excessive complexity. Other famous techniques include hybrid pre- and post-compensation [7], Volterra-based nonlinear equalisation (NLE) [8], phase-conjugated twin waves (PC-TW) [9], and the nonlinear Fourier transform (NFT) [10]. Unfortunately, pre-/post-compensation algorithms and Volterra-NLE present marginal performance

enhancement, PC-TW sacrifices signal capacity and NFT is unpractical for real-time signal processing. Above all, the aforementioned deterministic methods are unable to tackle stochastic nonlinearities, such as the amplified spontaneous emission (ASE) noise induced from optical amplifiers.

Unsupervised machine learning clustering has been recently introduced in optical communications for blind (training data-free) nonlinear equalisation (BNLE). Such unsupervised algorithms can tackle stochastic nonlinearities and include, for example, fuzzy logic C-means [11], K-means [11,12], hierarchical [11], affinity propagation [13], and Gaussian mixture [14] clustering. The algorithms used in [11–14] are mainly used for compensating fiber nonlinearity in long-haul transmission systems, in which the distorted received constellation diagrams contain Gaussian-circular clusters. However, there is an uncertainty if the machine learning clustering algorithms can be effective for non-circular rotated clusters caused by the strong nonlinear phase noise.

In this work, the aforementioned issue is addressed by experimentally demonstrating the first BNLE that harnesses the Density-Based Spatial Clustering of Applications with Noise (DBSCAN) [15] algorithm in 40 Gb/s 16 quadrature amplitude modulation (QAM) coherent optical signals being transmitted at 50 km. As a proof of concept, DBSCAN is tested for very high launched optical powers (LOPs), where the clusters of the received constellation diagrams are vastly rotated by means of self-phase modulation (SPM). Two modified DBSCAN methods are also adopted, in which the "un-clustered" noisy points are further processed using (1) K-means and (2) the minimum distance between an unlabelled point and the clustered points. It is shown that DBSCAN offers up to 0.83 dB Q-factor improvement over K-means and 8.84 dB when compared to linear equalisation at +16 dBm of LOP. This occurs because DBSCAN can effectively recover non-circularly-symmetric (elliptical form) noisy clusters by effectively combating SPM.

## 2. Density-Based Spatial Clustering of Applications with Noise (DBSCAN) Description

In density-based clustering, an assumption is made: clusters are dense regions in space, separated by regions of lower density [16]. A dense cluster is a region which is "density connected", i.e., the density of points in that region is greater than a minimum [17]. DBSCAN is an example that searches for dense areas and expands these recursively to find arbitrarily dense-shaped clusters. The two main parameters of DBSCAN are the $\varepsilon$ ("Epsilon") and the "minimum points". The $\varepsilon$ defines the radius of the "neighbourhood region" while the "minimum points" define the minimum number of constellation points (i.e., symbols) that should be contained within that neighbourhood. DBSCAN arbitrarily picks up a point until all of them have been visited. If the predefined number of "minimum points" is within the radius $\varepsilon$, then all these points are considered to be part of the same cluster. The clusters are then expanded by recursively repeating the neighbourhood calculation for each neighbouring point. However, for the unallocated points, if the number of points within the $\varepsilon$-neighbourhood is less than a predefined threshold, they are designated to be "noisy" and are not assigned to a particular cluster. Noisy data are not further processed in conventional DBSCAN. Here, a 2nd loop clustering is proposed to be applied only for these noisy data using (1) K-means [11] or (2) the minimum distance between an unlabelled point and the clustered points. A schematic diagram for conventional DBSCAN is depicted in Figure 1 when the minimum points are 4. In Figure 1, the following assumptions are assumed [18]:

a. Epsilon neighbourhood (N$\varepsilon$): A set of all constellation points within a distance $\varepsilon$.
b. Core point: A constellation point whose N$\varepsilon$ contains at least a "minimum point" (including itself).
c. Direct Density Reachable: A point $q$ is directly density reachable from a point $p$, if $p$ is a core point and $q \in$ N$\varepsilon$.
d. Density Reachable: Two constellation points ($p$, $t$) are density reachable if there is a chain of "direct density reachable" points that link these two points ($p$, $q$, $t$).
e. Border Point: A constellation point that is "direct density reachable" but not a core point.
f. Noise: Constellation points not belonging to any point's N$\varepsilon$.

The steps related to the conventional and modified DBSCAN are listed below, where the algorithm converges until all constellation points have been allocated to a cluster or labelled as "noisy" only if conventional DBSCAN is considered (step 5 below—1st loop) [17,18]:

1. Randomly select a point $p$ (referred in Figure 1) in the constellation map.
2. Retrieve all constellation points directly density reachable from $p$ that satisfy the condition of the radius $\varepsilon$ limits.
3. If the constellation point $p$ is a core point, a cluster is formed. Search recursively and find all of its density-connected points and assign them to the same cluster as $p$.
4. If $p$ is not a core point, the DBSCAN algorithm "scans" for the rest of the unvisited constellation points.
5. DBSCAN 1st loop: Points that are un-clustered are labelled as zero points ("noisy points") where linear equalisation is performed only on these points, and then the conventional DBSCAN algorithm stops.
6. DBSCAN 2nd loop (extra novel step):

   i. *Method-1:* K-means clustering is activated for the "noisy points" using the Lloyd's algorithm [11,12]:

a. Assignment: Allocate each observation to the cluster whose mean has the least squared Euclidean distance ("nearest" mean) [11,12].
b. Update: Calculate the new means to be the centroids of the observations in the new clusters [8,9]. K-means converges when assignments do not change.

   ii. *Method-2:* Calculation of minimum distance between the unlabelled "noisy points" and the clustered points.

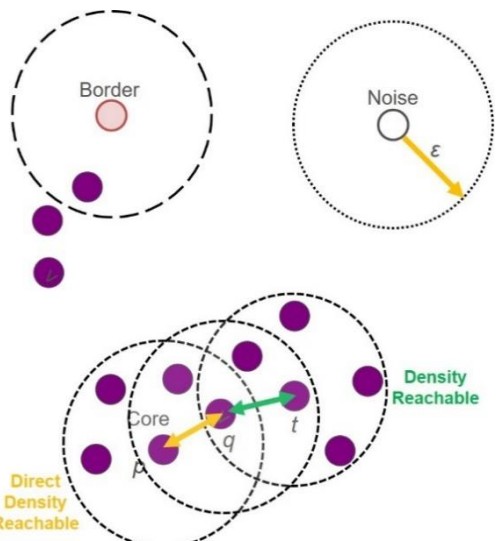

**Figure 1.** Density-Based Spatial Clustering of Applications with Noise (DBSCAN) example for Min. Points = 4.

## 3. Experimental Setup

Figure 2 depicts the schematic diagram of the experimental setup of the 10 GBaud (40 Gb/s) 16 QAM coherent signal. In the transmitter-digital signal processing (DSP), look-up table-based pre-distortion was used to mitigate the opto-electronic component impairments, similarly to [19]. A narrow linewidth (<100 kHz) external cavity laser (ECL) was tuned to 1549.5 nm and, using an arbitrary waveform generator (AWG) operating at 20 GS/s, two uncorrelated pseudo-random level

signals ($2^{15}-1$) were applied to the in-phase/quadrature (I/Q) modulator to generate the 16 QAM signal. After IQ modulation, the optical signal was transmitted over 50 km of standard single-mode fiber (SSMF). At the receiver, noise loading was added using an optical amplifier to set different optical signal-to-noise ratio (OSNR) values and subsequently the optical signal was converted to an electrical one using a homodyne coherent receiver. Afterwards, the signal was captured by a real-time oscilloscope sampled at 50 GS/s for offline receiver-DSP, in which the data was first resampled to two samples/points using prior knowledge of the clock frequency. Then the constant modulus algorithm (CMA) combined with the multi-modulus algorithm (MMA) was utilised for signal equalisation. An $M^{th}$ power frequency drifting compensation method was employed to compensate the frequency offset between the signal and the local oscillator in the coherent receiver. The decision-directed phase-locked loop (DDPLL) method was employed for the carrier phase recovery. Finally, the machine learning algorithm was processed before the hard decision and the bit error rate (BER)/Q-factor ($=20\log_{10}\left[\sqrt{2}erfc^{-1}(2BER)\right]$) calculation, similarly to other reported work with machine learning signal processing [20–25].

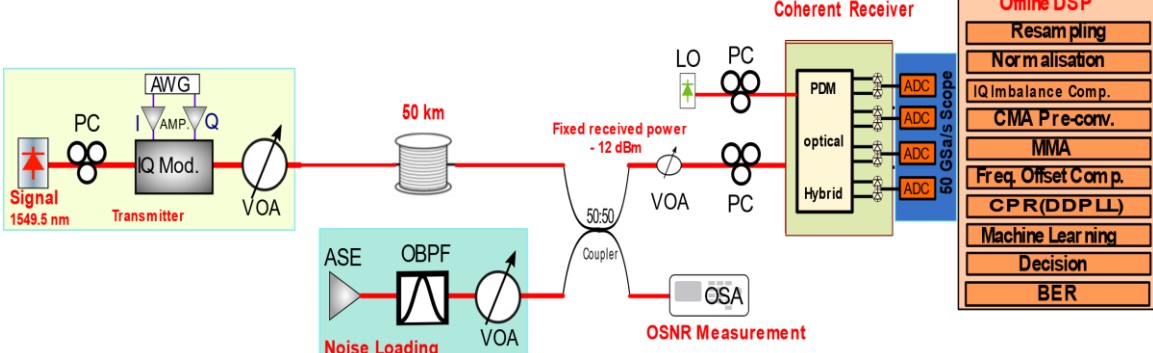

**Figure 2.** Experimental setup for a 40 Gb/s 16 quadrature amplitude modulation (QAM) coherent optical signal transmitted at 50 km, incorporating machine learning clustering. PC: polarisation controller, OBPF: optical band-pass filter, LO: local oscillator, CMA/MMA: constant/multi-modulus algorithm, CPR (DDPLL): carrier phase recovery (decision-directed phase-locked loop).

## 4. Results

A 40 Gb/s 16 QAM waveform is transmitted with +16 dBm LOP over 50 km. Two parameters are needed to optimise the DBSCAN algorithm to produce the lowest BER, namely $\varepsilon$ and the minimum number of points. It is worth noting that in a real-time communication link, the optimisation process of DBSCAN should be run offline as a training stage. In this stage, the optimal Voronoi regions would be generated, where in the real-time processing the incoming data would be assigned directly to ideal symbols. It is envisaged that such an approach should be very effective in optical communication links, where linear and nonlinear effects remain relatively stable over time, and therefore the training process should be run only once. The calculated BER while scanning for $\varepsilon$ and the minimum number of points is shown in Figure 3. From Figure 3, it is evident that the majority of the best BER values can be found for $0.1 < \varepsilon < 0.45$. For these values of $\varepsilon$, the minimum points do not affect the BER much except when $0.1 < \varepsilon < 0.14$, where the minimum points should be less than 110. In Figure 4, the performance of clustering algorithms is shown for different LOPs and two values of received OSNR: 30 and 15 dB. In Figure 4, the performance benefit of machine learning clustering over linear equalisation is significant for both OSNR values, especially when using DBSCAN method-2, resulting in up to 8.8 dB Q-factor improvement. This is attributed to the compensation of SPM since single-channel transmission is carried out. Results indicate that DBSCAN-based BNLE is a robust soft-clustering method when very strong nonlinear phase noise is present and where linear equalisation fails completely. Moreover, DBSCAN method-2 has the highest Q-factor along the whole range of LOPs. Compared to DBSCAN method-1 and K-means, method-2 increases the Q-factor by up to about 0.7 and 0.83 dB, respectively,

by better handling highly rotated clusters that become almost elliptically shaped. This is because the overlapping (soft) clustering ability of method-2 is more powerful than the common hard (exclusive) clustering of K-means and method-2 (which also includes K-means for the noisy constellation points). This is confirmed by the received constellation diagrams of Figure 5b, related to +16 dBm of LOP (OSNR = 30 dB). On the other hand, the quite similar performances between DBSCAN algorithms and K-means at lower LOPs and OSNR is due to the existence of nearly circular Gaussian clusters, which are not rotated. This is corroborated in the received 16 QAM constellation diagrams of Figure 5a at +10 dBm of LOP (OSNR = 15 dB). In the left constellation diagram of Figure 5a, the DBSCAN "noisy" points are also presented in the 1st loop of the algorithm.

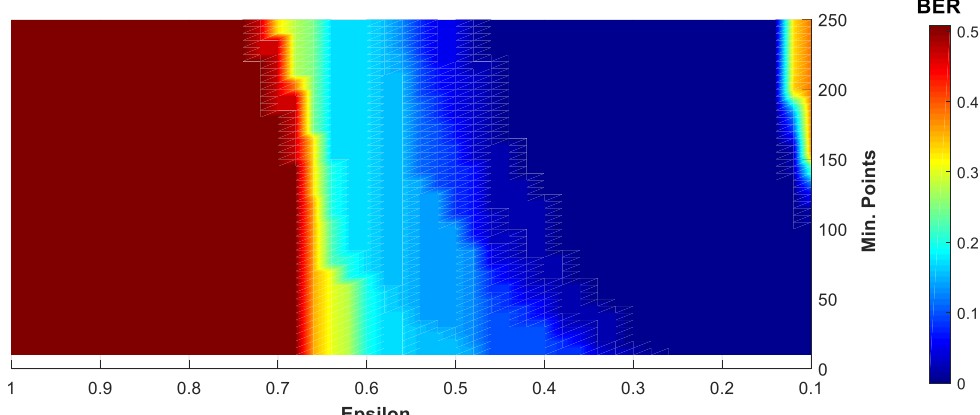

**Figure 3.** DBSCAN optimisation for 16 QAM transmission over 50 km at +10 dBm of launch power: bit-error-rate (BER) vs. $\varepsilon$ ("Epsilon"), Min. Points = 4.

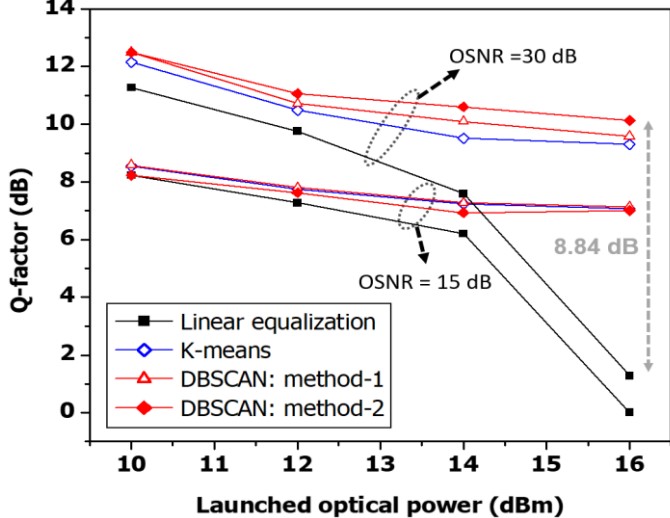

**Figure 4.** DBSCAN vs. K-means for 16 QAM transmission at 50 km for different launched optical powers (LOPs) when optical signal-to-noise ratio (OSNR) is 30, 15 dB.

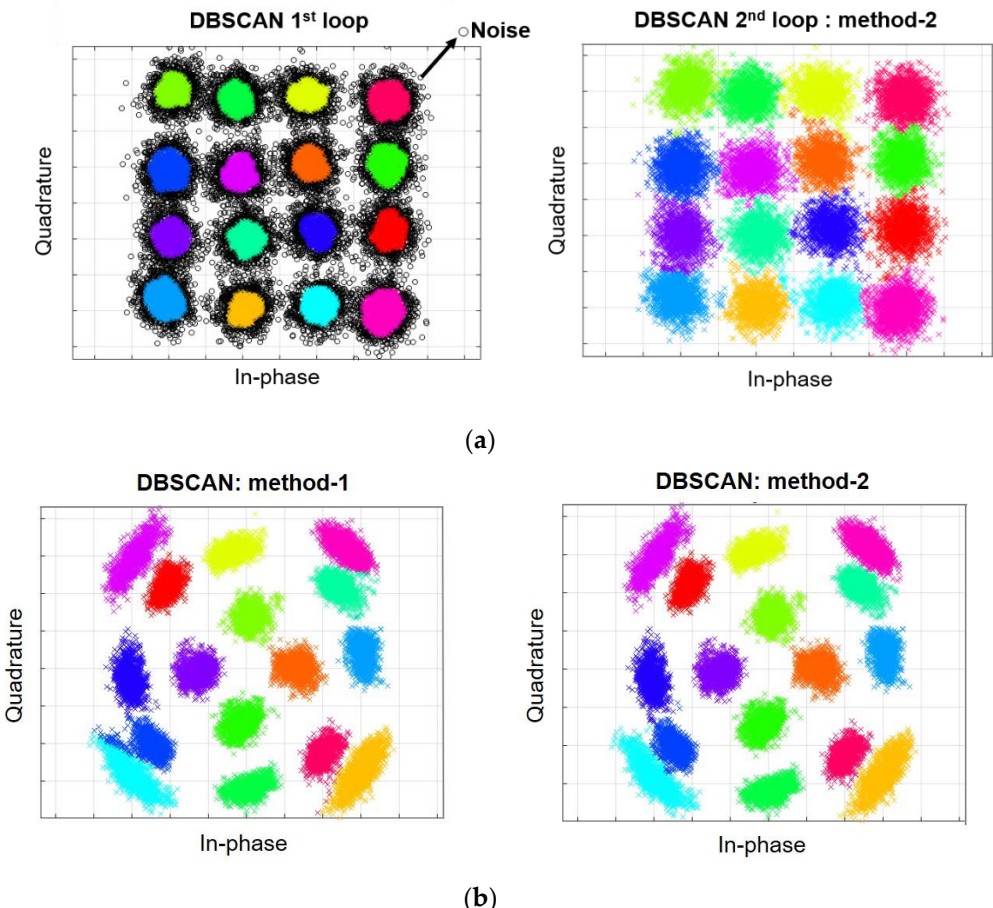

**Figure 5.** Received constellation diagrams for (**a**) DBSCAN 1st loop (left), method-2/2nd loop (right) at +10 dBm of LOP (OSNR = 15 dB); and (**b**) DBSCAN method-1 (left), method-2 (right) at +16 dBm of LOP (OSNR = 30 dB).

## 5. Conclusions

The first DBSCAN-BNLE was experimentally demonstrated for 16 QAM at 50 km. Two novel DBSCAN methods were proposed, in which the "un-clustered" noisy constellation points were processed using (1) K-means and (2) the minimum distance between an unlabelled point and the clustered points. Compared to linear equalisation, method-2 improved the Q-factor up to 8.8 dB by combating SPM. Method-2's ability for overlapping clustering resulted in Q-factor improvement over method-1 and K-means (exclusive clustering), when vastly rotated clusters of nearly elliptical form occur. Once optimised, DBSCAN proved to be a robust BNLE for very strong nonlinear phase noise.

In future work, inter-channel nonlinear effects, such as four-wave mixing and cross-phase modulation, will be tackled using the proposed machine learning algorithm in a super-channel transmission scenario that incorporates many wavelength channels.

**Author Contributions:** Conceptualization, E.G. and Y.L.; methodology, E.G.; software, E.G. and Y.L.; validation, M.J., S.O., K.M., P.F.W. and L.P.B; formal analysis, M.J.; investigation, E.G.; resources, L.P.B.; data curation, K.M. and P.F.W.; writing—original draft preparation, E.G. and Y.L.; writing—review and editing, E.G., Y.L., S.O., and L.P.B; visualization, E.G., Y.L. and S.O.; supervision, L.P.B.; project administration, E.G. and L.P.B.; funding acquisition, E.G. and L.P.B.

**Funding:** This work was supported by the Science Foundation Ireland through grant numbers 13/RC/2077, 12/RC/2276, 15/US-C2C/I3132, the HEA INSPIRE Programme, and the EU/EDGE Marie Curie programme with grant number 713567.

**Conflicts of Interest:** The authors declare no conflict of interest. The funders had no role in the design of the study; in the collection, analyses, or interpretation of data; in the writing of the manuscript, nor in the decision to publish the results.

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
