# Peer review of "A Blind Nonlinearity Compensator Using DBSCAN Clustering for Coherent Optical Transmission Systems"

_applsci, doi:10.3390/app9204398_

Round 1
Reviewer 1 Report
The paper addresses machine learning method for nonlinear mitigation in coherent optical communications. I think it is interesting in general but in detail i have few questions and comments which the paper can be published after a few clarifications.
1- P.2 L.44 the sentence starts with "However,..." is too long and is hard to understand (at least for me). please correct it.
2- In the experimental setup it seems that authors transmit data in only 1 polarization while in practice usually 2 polarization is used (polarization multiplexing). What would be the effect of PM in your approach? why have not you experimented with PM?
3- In figure (1), i think it would be better if you show the chain of points you mentioned in the text for "Density Reachable" point.
4- In figure (3), it seems that the BER is independent of minimum points!! at least the figure does not show any dependency. please provide a figure to show the dependency or explain in the text about dependency to minimum points. I suggest to draw 2-dim figures and not 3-dim figures that are not readable.
5- In your experiment you only considered one channel. So you only consider Self Channel Interference (SCI). In WDM, we certainly have other channels interference (XCI). what would be the effect of XCI in your approach? Why did not you experiment WDM ?
6- page.4 line 141 0.45<epsilon<0.1 is a mistake 0.1<0.45.
7- Is the epsilon and minimum points optimum range that you mentioned in 141-142 works for general PM-WDM scenarios(different number of channels and different modulation formats and different fiber types)? or they must be calculated for each scenario? How can we have a theoretical estimation in practice ?
8- In figure (4) you used Q-factor but it is not defined mathematically. It would be better to mention its definition in the text.
9- The caption of figure (5) is in page.6 . It would be better that figure and its caption to be in the same page.
Reviewer 2 Report
The abstract is not well-written. Authors should lead with the problem they are providing a solution for.
Parts of the introduction needs proper citation.
The writing style should be changed to a passive voice. The experimental setup section is the only correct section written in a passive voice.
The graphics included in the figures are of low quality. Higher quality figures are required.
The complexity of the system reported in the conclusions section is unjustified. The mention of including a detailed analysis in a future publication, simply does not suffice.

Round 2
Reviewer 2 Report
Thank you for improving the transcript. Just a minor improvement for figure one, the labels and circles overlap.
Author Response
We would like to thank the reviewer for notifying the figure error.
In the final version we have included a new corrected figure.
